# Chondroitin-Sulfate-A-Coated Magnetite Nanoparticles: Synthesis, Characterization and Testing to Predict Their Colloidal Behavior in Biological Milieu

**DOI:** 10.3390/ijms20174096

**Published:** 2019-08-22

**Authors:** Ildikó Y. Tóth, Erzsébet Illés, Márta Szekeres, István Zupkó, Rodica Turcu, Etelka Tombácz

**Affiliations:** 1Department of Applied and Environmental Chemistry, University of Szeged, Rerrich Béla tér 1, H-6720 Szeged, Hungary; 2Department of Physical Chemistry and Materials Science, University of Szeged, Rerrich Béla tér 1, H-6720 Szeged, Hungary; 3Department of Pharmacodynamics and Biopharmacy, University of Szeged, Eötvös u. 1, H-6720 Szeged, Hungary; 4National Institute R&D for Isotopic and Molecular Technology, Donat Street 67-103, 400293 Cluj-Napoca, Romania; 5Department of Food Engineering, Faculty of Engineering, University of Szeged, Moszkvai krt. 5-7, H-6725 Szeged, Hungary

**Keywords:** magnetite, chondroitin-sulfate, adsorption, core-shell nanoparticles, surface complexation, surface spectroscopy, chemical stability, coagulation kinetics, colloidal stability, MTT assays

## Abstract

Biopolymer coated magnetite nanoparticles (MNPs) are suitable to fabricate biocompatible magnetic fluid (MF). Their comprehensive characterization, however, is a necessary step to assess whether bioapplications are feasible before expensive in vitro and in vivo tests. The MNPs were prepared by co-precipitation, and after careful purification, they were coated by chondroitin-sulfate-A (CSA). CSA exhibits high affinity adsorption to MNPs (H-type isotherm). We could only make stable MF of CSA coated MNPs (CSA@MNPs) under accurate conditions. The CSA@MNP was characterized by TEM (size ~10 nm) and VSM (saturation magnetization ~57 emu/g). Inner-sphere metal–carboxylate complex formation between CSA and MNP was proved by FTIR-ATR and XPS. Electrophoresis and DLS measurements show that the CSA@MNPs at CSA-loading > 0.2 mmol/g were stable at pH > 4. The salt tolerance of the product improved up to ~0.5 M NaCl at pH~6.3. Under favorable redox conditions, no iron leaching from the magnetic core was detected by ICP measurements. Thus, the characterization predicts both chemical and colloidal stability of CSA@MNPs in biological milieu regarding its pH and salt concentration. MTT assays showed no significant impact of CSA@MNP on the proliferation of A431 cells. According to these facts, the CSA@MNPs have a great potential in biocompatible MF preparation for medical applications.

## 1. Introduction

Superparamagnetic nanoparticles (e.g., maghemite, magnetite, and cobalt-ferrite), usually in organic solvent or water based ferrofluid formulations, are among the most promising nanoparticle systems for both engineering and biomedical applications [1,2,3,4,5,6,7,8,9,10,11,12,13,14]. The organic solvent based magnetic fluids (MFs) are used in industrial and economic applications, e.g., in high performance HiFi systems [12], hard disks [13], or seals operating under extreme conditions [8,14]. The water based magnetic fluids are planned for biomedical applications in modern medicine in both diagnostics and in therapy, such as magnetic resonance imaging (contrast agent in MRI), magnetic particle imaging, targeted drug delivery, and magnetic hyperthermia therapy [2,3,4,5,6,7,9,10,11,15,16,17]. These in vivo applications require that MFs are non-toxic and that the magnetic particles are sufficiently uniform in size, well-dispersed in aqueous media, and chemically and colloidally stable under in vivo conditions (e.g., in blood: pH ~7.2–7.4, ~150 mM salt concentration and the presence of proteins).

To fulfill these criteria, the magnetic nanoparticles must be coated [18,19], for example, by small organic molecules (e.g., citric acid [10,20,21,22] and oleic acids [23,24]), neutral polymers (e.g., natural dextran [20,25], synthetic polyethylene-glycol [20,25,26,27]), or by polyelectrolytes (e.g., synthetic polyacrylic acid (PAA) [20,21,22]). The enhanced colloidal stability of magnetic nanoparticles can be achieved by electrostatic (e.g., citric acid coating), steric (e.g., dextran coating) or electrosteric stabilization (e.g., polyacrylic acid coating) [28,29]. The thicker layer of a polyelectrolyte shell provides better electrosteric stability as compared to the stabilization by the adsorbed small molecules of citric acid dissociated at pH > 4 [21].

Usually, the non-toxic magnetite nanoparticles (MNPs) are used to prepare biocompatible magnetic fluids. There are two typical ways to coat MNPs [20]. One of these is the in situ coatings where the nanoparticles are coated during the precipitation of iron oxide [20,30,31,32,33]. The other is the post-synthesis coating method, in which the surface of the previously synthesized and purified magnetic particles are modified [19,20,21,22].

Polysaccharides are the most promising natural stabilizing agents for magnetic nanoparticles [34], e.g., dextran-coated iron oxide (Ferumoxides and Ferumoxtran-10) and carbo-dextran-coated iron oxide (Ferucarbotran) are well known products for biomedical applications [22,35]; furthermore, chitosan [36,37] and chondroitin-sulfate (CS) [38] with polyelectrolyte character are suitable for this purpose too. 

Chondroitin-sulfate (CS) is a sulfated natural polysaccharide. One repeating unit of CS is comprised of a glucuronic acid and an N-acetyl-galactoseamine ring, the latter modified by sulfate groups replacing –OH groups. There are several types of CS depending on the positions and the quantities of the –SO_3_H groups, chondroitin-sulfate-A (CSA, chondroitin-4-sulfate), and chondroitin-sulfate-C (CSC, chondroitin-6-sulfate) being the most common types [39]. One repeating unit of CSA contains two dissociating groups, a –COOH and a –SO_3_H group, with significantly different protonation/deprotonation properties. The strongly acidic sulfate groups are fully deprotonated in a wide pH-range [40,41], while the –COOH groups are characterized by pH-dependent dissociation with pK_β-glucuronic acid_ ~2.9 [42]. Regarding the biomedical usability, chondroitin sulfate proteoglycans play a key role in tumor growth, since the negatively charged CS chains interact with a large number of ligands and receptors, and activate signaling pathways, which stimulate tumor growth. However, some modified CSs such as oversulfated CS are known as potential anticancer agents [43]. It is highly probable that CSA coated MNPs are able to accumulate near the tumor, thereby efficiently acting as a contrast agent in MRI and locally in magnetic hyperthermia.

The chondroitin-sulfate iron colloid complex is a therapeutic agent for iron deficiency anemia (Blutal, Dainippon Pharmaceutical Co.), which has been marketed for more than 30 years [44,45]. There are patents for magnetic nanoparticles prepared in the presence of CS for a feasible MRI contrast agent [33,46], and several other publications can be found on magnetite coated by chondroitin-sulfate [30,47,48]. Novel CSA-coated core–shell magnetite nanoparticles (CSA@MNP) were recently prepared successfully by a well-defined post-synthesis coating method [49]. The resulting magnetic fluid has had sufficient colloidal stability for in vivo applications, but its exact physical, physicochemical, and colloidal properties have not been explored comprehensively. The qualitative and quantitative characterization of shell formation on the magnetic core; the determination of the bond formation between CSA and MNP surface; the quantitative optimization of coating to stabilize MNPs efficiently; the characterization of core-passivation efficiency; the determination of particle charge; and the salt tolerance measurements of CSA-coated MNPs are effective and inexpensive tools to optimize the MNP coating process, and to prove the feasibility of biological applicability of the prepared materials [22].

The fundamental aim of this work is the comprehensive characterization of CSA@MNPs prepared by a post-synthesis coating method, and designed for biomedical applications. First, we synthesized the magnetic core by co-precipitation, and clarified the composition of CSA by TG and potentiometric acid-base titration. Then, we studied quantitatively the adsorption of CSA on MNPs by analyzing the adsorption isotherm, and characterized the CSA@MNP particles by TEM and VSM techniques. Next, we identified the chemical bonds between the MNPs and CSA by FTIR-ATR and XPS spectroscopies. After that, we studied the pH-dependent surface charging and aggregation properties of the CSA@MNP particles, and tested the capability of magnetic fluids for biomedical applications by coagulation kinetics studies, and also measured the dissolved iron-content responsible for the damage caused by ROS in vivo. Finally, we investigated the toxicity of the optimized CSA@MNPs in cytotoxicity experiments. We would like to emphasize that the characteristic material properties investigated here are very closely related to possible biological applications. Nel et al. [50] have demonstrated that different propensities of bionanointerfaces such as surface charge hydrophobicity (dispersibility) of particles correlate well with the results of toxicity tests in biological systems. Our studies on MNPs’ hemacompatibility [51] have proved that colloidal stability testing at biorelevant pH and salty conditions can reveal blood incompatibility, that ‘promises’ fatal outcomes in vivo such as thrombosis in the case of intravenous administration.

## 2. Results and Discussion

### 2.1. Adsorption of CSA on MNP 

The adsorption isotherm of CSA on MNP determined by the batch method using both density measurement and enzymatic digestion followed by UV-absorption concentration determination is presented in Figure 1. The detection limit of the well-known enzymatic digestion measurements was ~1 mmol/L, so density measurements were used to determine the adsorption isotherm at equilibrium CSA concentrations lower than that. The two different series of results fit well together. 

The point of zero charge (PZC) of naked MNPs is at pH ~8.0. At pHs lower than the PZC, the surface charge of MNP is positive due to the presence of ≡Fe–OH_2_^+^ groups on the surface, while the particles are negatively charged above the pH of PZC because of the ≡Fe–O^−^ groups [52]. At pH ~6.3 and 10 mM NaCl, the amount of the positive charge on MNPs is ~0.05 mmol/g from acid-base titration [19]. It is not enough to electrostatically stabilize the particles; therefore, the naked magnetite is aggregated and settled, as seen in the first vial of Figure 1. Above ~0.2 mmol/g of CSA-addition, the CSA-coated magnetite nanoparticles become well dispersed (see in the third vial on Figure 1) due to the electrosteric stabilization effect of the adsorbed polyelectrolyte (see the characterization of CSA dissociation in Appendix A). 

CSA has an H-type (high-affinity) adsorption isotherm on the MNP surface with a high-affinity limit of adsorption at ~0.06 mmol/g (close to the amount of the positive charge on MNP ~0.05 mmol/g) and the plateau is reached at ~0.10 mmol/g at relatively low equilibrium concentration. These values express mmoles of the CSA repeating unit, so the amount of the negative charges carried by adsorbed CSA is about twice of them, since the total amount of negative charges at the pH and ionic strength of the experiments is 1.9 mol/mol of repeating units (see Appendix A for details). The shape of the CSA isotherm is similar to that of poly(acrylic acid-co-maleic acid) copolymer (PAM) at the same pH and ionic strength [19], but the adsorbed amounts are much smaller (the high-affinity limit is at ~0.3 mmol/g, and the plateau is at ~0.9 mmol/g for the PAM adsorption). The large differences can be explained by the molecular structure of CSA. The charge density (i.e., the amount of negatively charged functional groups on the unit length of the chain) can be estimated from the molecular structure and the pH- and ionic strength-dependent charging of the polyelectrolytes. At pH ~6.3 and 10 mM NaCl, the calculated charge density of the CSA chain is about –2.1 nm^−1^ (i.e., there are 0.95 –SO_3_^‒^ and 0.95 –COO^‒^ groups corresponding to the ~0.9 nm long repeating unit [53]), which is much smaller than the ~ –4.5 nm^−1^ characteristic for PAM (i.e., the repeating unit of PAM is ~0.4 nm long and it contains 3 –COOH groups, the degree of dissociation of which is 0.6 [19]). The molecular structure of CSA is even more complex; it contains several potential donor groups serving as ligands for adsorption such as carboxylate, sulfate, alcoholic hydroxyl, or amide groups [54,55,56,57]. In addition, the chain of CSA is much less flexible than that of PAM, which together with the relatively wide ring structure of the CSA backbone gives rise to a significant steric constraint of the adsorption. 

In the photos of CSA@MNP samples (inserted in Figure 1), the signs of aggregation (large particles on the glass surface and partial sedimentation) can be seen even at high equilibrium concentration in the range of the adsorption plateau. The CSA prepared from bovine trachea contains a small amount of cetylpyridinium chloride (CPCl) contamination [58,59], which can be adsorbed on MNP resulting in the partial hydrophobization of the surface and the aggregation of CSA@MNP samples. This was proved by UV-measurements. The characteristic absorption of CPCl at ~260 nm [59,60,61] was detected in the solution of CSA, and it was absent in the equilibrium supernatant of CSA@MNP samples even at high CSA concentration. The presence of cetylpyridinium chloride is not favorable for preparing stable CSA@MNP samples, but this problem can be solved by production under accurate conditions (specific pH, ionic strength, and ultrasonication).

### 2.2. Size and Magnetic Property of Magnetic Core in CSA@MNP

These propensities of MNPs’ product for biomedical use are crucial because excretion of NPs depends on their size, and their efficacy in MRI or hyperthermia is based on the magnetic character. The TEM image and the specific magnetization curve of the stable CSA@MNP particles are presented in Figure 2. The TEM image of the naked MNP below the coated sample is also shown for direct comparison. Due to the presence of surface coating that prevents the close aggregation of nanoparticles [62], many white gaps can be identified among MNPs in the TEM image of CSA@MNP sample (Figure 2a top), especially compared directly to the naked one (Figure 2a bottom). The difference in the degree of aggregation is clearly visible in the TEM images. The aggregates containing mostly the larger primary particles are probably present in the suspension of the naked sample, which are dispersed because of the stabilizing effect of CSA-coating. The latter is also supported by a limited increase in the average particle size of the CSA@MNP sample related to that of the naked one, as shown in the insets of TEM images in Figure 2a. It should be noted that only discrete particles can be taken into account when determining average size distribution.

The magnetization curve of CSA@MNPs in fluid media (Figure 2b) shows superparamagnetic behavior at room temperature, as it is expected for magnetite nanoparticles of small sizes, of about 10 nm that are surface coated by a non-magnetic organic layer. The measured magnetization values were converted to specific magnetization, using the exact amount of magnetite in the sample. The mass of the CSA and medium was not involved in this calculation. The saturation magnetization (M_S_) was obtained by extrapolation to an infinite field in an M vs. 1/H^2^ plot [63]. Thus, the M_S_ of the superparamagnetic CSA@MNPs is 56.9 emu/g related to the exact amount of magnetite. This value is in the range of 53–58 emu/g measured for the polyacrylic acid-coated MNPs, and suggests no decrease in the size of the magnetic core due to iron dissolution on CSA adsorption, contrary to that observed in citrated MNPs (~42 emu/g at high citric acid-loading) [22].

### 2.3. Binding of CSA on MNPs 

FTIR-ATR analysis was used to characterize the bond formation on ≡Fe‒OH sites. Figure 3 shows the IR absorption spectra of pure MNP, CSA, and CSA-coated MNP. The positions of characteristic bands are collected in Table 1.

In the spectrum of pure MNP, two characteristic bands are present, i.e., the H-bonded OH-stretching at ~1620 cm^−1^ [64] and the Fe−O band at ~548 cm^−1^ [65,66]. The Fe−O band shifts to higher wavenumber (~559 cm^−1^) upon CSA adsorption.

The characteristic bands of deprotonated sulfate groups (–O–SO_3_^−^) in the spectrum of CSA are the S = O and C−O−S stretching vibrations at ~1260 cm^−1^ and ~856 cm^−1^, respectively [57,58,67,68]. The position of the C−O−S vibration confirms that the current CS sample is almost completely CSA [58]. The characteristic bands of –O–SO_3_^−^ remained unchanged during the adsorption, so the sulfate groups are not involved in the adsorption [57]. The vibration of amide groups at ~1560 cm^−1^ [57,64,67] as well as the vibrations of the alcoholic hydroxyl groups and pyranose rings at 1234 cm^−1^, 1414 cm^−1^, 1055 cm^−1^, 1035 cm^−1^, and 925 cm^−1^ [67] do not change during the adsorption, so these groups are also not able to bind on the magnetite surface.

The characteristic band of protonated carboxyl groups (−COOH) is the C = O stretching band around ~1740 cm^−1^ [64,69], which does not appear in the spectrum of CSA. However, the deprotonated carboxylates provide the asymmetric and symmetric vibrations of the −COO^−^ resonant structure around ~1612 cm^−1^ and ~1379 cm^−1^ [64,67,68,69]. These data indicate that the carboxyl groups of CSA are almost completely deprotonated (−COO^−^) at pH ~6.3 and 10 mM NaCl. This is also expected on the basis of titration results (Appendix A), namely that the amount of protonated carboxylates at pH ~6.3 and 10 mM NaCl is negligible: 0.05 mol/mol of repeating units.

The C = O stretching band does not appear even in the spectrum of CSA-coated MNP, so the protonated –COOH groups are not involved in the mechanism of adsorption. Therefore, there is no H-bond formation between the CSA and MNP. However, the asymmetric and symmetric vibrations of C−O in the COO^−^ groups are shifted significantly (see Δν values in Table 1) after adsorption, indicating the formation of direct metal-carboxylate complexes [19,69]. Both the neutral (≡Fe–OH) and positively charged (≡Fe–OH_2_^+^) surface groups of MNPs can be involved in the formation of this inner-sphere metal–carboxylate complexes: ≡Fe–OH + ^−^OOC–R → ≡Fe–OOC–R + OH^−^(1)
≡Fe–OH_2_^+^ + ^−^OOC–R → ≡Fe–OOC–R + H_2_O(2)
This type of complex-formation can explain the H-type isotherm of CSA adsorption on the MNP.

The high resolution XPS spectra of Fe, C, O, N, and S in the CSA@MNP sample are shown in Figure 4. The peaks were deconvoluted into the components. The fitting parameters of the deconvolution, i.e., the peak positions (binding energy) and the full width at half-maximum (fwhm) together with the atomic concentrations calculated from the peak areas are given in Table 2. In bulk magnetite (Fe(III)_2_Fe(II)O_4_), the theoretical Fe^3+^/Fe^2+^ atomic ratio is 2.0. However, this value decreased to 1.4 in the case of CSA-coated MNPs, as calculated from the Fe^3+^ and Fe^2+^ 2p peaks, which shows the reduction of iron at the MNP surface in the presence of CSA polysaccharide. 

The chemical structure of the CSA is very complex, and thus the assignment of the electron binding energies determined from the deconvolution of the XPS peaks belonging to the moieties of adsorbed CSA is difficult. The N 1s peak and the S 2p peaks were found at 399.67 eV, 169.95 eV, and 166.77 eV, respectively. The best fit for C 1s spectrum was obtained with four components: The peaks at 284.61 eV and 285.90 eV were assigned to C‒C, C‒H and C‒O, C‒N, respectively; the peaks at 288.29 eV and 289.48 eV were assigned to N‒C = O, C‒O(‒SO_3_^‒^) and O‒C = O. The ratio between the moieties with lower (284.61 eV and 285.90 eV) and higher (288.29 eV and 289.48 eV) binding energies calculated from the atomic concentrations of C 1s peaks is 4.1 (Table 2), while the same ratio calculated from the molecular structure of CSA is 3.7. The O 1s peak at 530.51 eV was assigned to C‒O(‒C), C‒O(‒H), and C‒O(‒S), and that at 532.43 eV to O‒C = O, N‒C = O, and ‒SO_3_^‒^. The deconvoluted peak at 536.11 eV is the sign of adsorbed water [62,70,71]. The ratio between the atomic groups with lower (530.51 eV) and higher (532.43 eV) binding energy calculated from the atomic concentrations of O 1s peaks is 1.49, while that calculated from the molecular structure of CSA is 1.37. The results show that the relative amount of the bonds with higher electron binding energies is smaller in the adsorbed CSA. This observation also confirms the formation of inner-sphere metal–carboxylate complex.

### 2.4. The Effect of CSA Adsorption on the Charge State and Aggregation of MNPs 

The electrokinetic potential of the nanoparticles and the adsorbed amount of CSA on MNPs at pH ~6.3 and 10 mM NaCl are plotted as a function of CSA-loading in Figure 5. As was already discussed in the adsorption of CSA on MNP section, the MNPs are positively charged and settled under this condition, as seen in the first picture inserted in Figure 5. The electrokinetic potential of naked MNPs is ~ +25 mV, and this value decreases with the increasing amount of added CSA carrying negative charges. The charge neutralization occurs at ~0.035 mmol/g of CSA-loading, and the further addition of CSA causes charge reversal, and increases the absolute value of the negative electrokinetic potential. The CSA@MNP particles become fully dispersed at >~0.2 mmol/g of CSA addition (see the pictures inserted in Figure 5) due to the electrosteric stabilization supported by the large absolute values of the electrokinetic potential, −40 mV similarly to that published before [49]. 

The isoelectric point (IEP, at which the net charge of CSA@MNP is zero) is at ~0.035 mmol/g of added CSA, which is lower than the high-affinity limit of adsorption (~0.06 mmol/g in Figure 1) meaning that practically all added CSA is adsorbed from the equilibrium solution phase. According to the degree of sulfonation (0.95 determined by TG Appendix A) and degrees of dissociations (~1.0 for –SO_3_H and ~0.95 for –COOH groups determined by potentiometric acid-base titration Appendix A) one mole of repeating units of CSA contains 0.95 mol –SO_3_^−^ and 0.95 mol -COO^‒^ groups under this condition. Thus, at the IEP, the total amount of charges on adsorbed CSA related to MNP is ~–0.067 mmol/g. This adsorbed negative charge slightly overcompensates the ~0.05 mmol/g value of the original positive charge of MNPs [19]. Under the electroneutrality condition of the isoelectric point, the excess negative charge introduced by CSA adsorption is neutralized by definition. The FTIR spectra showed the formation of direct metal-carboxylate bonds (Equations (1) and (2)), and the electrokinetic potential results also show that both the positively charged surface groups (Equation (2)) and the neutral surface sites (Equation (1)) are simultaneously involved in the formation of the inner-sphere metal–carboxylate complexes. According to the reaction of Equation (1), part of the CSA is neutralized in the course of adsorption without decreasing the original positive charge of the MNP surface, the latter process being responsible for the apparent charge overcompensation. 

The pH-dependent colloidal stability of MNPs with increasing CSA-loadings was monitored by the measurements of average hydrodynamic diameter and electrokinetic potential at 10 mM NaCl. The IEP values at different CSA-loadings and the pH ranges of nanoparticle aggregation have been collected in Table 3. Without CSA, the IEP is at pH ~8, and the dispersions of naked magnetite are only colloidally stable at pHs below 5 or above 10. By increasing CSA addition, the IEP shifts gradually to more acidic pH. At trace CSA-loading (0.05 mmol/g), the CSA@MNP particles aggregate in the whole pH-range studied here due to the patch-wise adsorption of oppositely charged CSA on MNP. A similar observation was made in humic acid adsorption on MNPs [52]. Further CSA addition leads to narrowing the pH-range of aggregation around the IEP, and the CSA@MNP dispersions become stabilized in the whole pH-range studied at loadings above ~0.2 mmol/g [49].

### 2.5. Salt Induced Aggregation of CSA@MNP at pH ~6.3 

The salt tolerance of naked and CSA-coated MNPs was characterized by stability ratios (W) determined at different NaCl concentrations in coagulation kinetics experiments at near physiological pH ~6.3. The stability plot (the logarithm of the stability ratio (log_10_ W) as a function of the logarithm of salt concentration (log_10_ c_NaCl_)) is shown in Figure 6. The dashed line refers to our previous results for PAM-coated MNPs [19] added here for comparison. 

The dispersion’s salt tolerance, i.e., the smallest amount of salt inducing fast coagulation (log_10_ W = 0), is represented by the values of critical coagulation concentration (CCC) [49,72,73]. The CCC determined as the intercept of the straight line of the slow coagulation regime (log_10_ W > 0) with the *x*-axis shifts gradually from ~1 mM for naked MNP to ~80, ~150, and ~500 mM at CSA-loading 0.1, 0.2, and 1.0 mmol/g, respectively. Details are given in the Appendix A.

The resistance against physiological concentration (CCC > 150 mM) is achieved at the CSA-addition of ~0.2 mmol/g corresponding to the adsorbed amount of CSA close to the saturation level (~0.11 mmol/g). Thus, the equilibrium concentration of the non-adsorbed CSA is low (~1 mmol/L), which is advantageous for possible bioapplication in the future. The salt tolerance of CSA@MNP particles reaches maximum (~500 mM) at ~1.0 mmol/g of added CSA.

### 2.6. Chemical Stability of MNP Coated with CSA 

Besides the redox transformation of magnetite to maghemite during storage, iron leaching may occur from any MNP products designed for biomedical applications. The exclusion of the leaching is very important, since a high amount of dissolved iron ions can hurt the living systems through oxidative stress [74,75,76]. The chemical stability of CSA@MNP systems was tested in iron dissolution experiments. The dissolved iron was determined in the equilibrium supernatant of aqueous dispersions in 10 mM NaCl solution at pH ~6.3 after one day of adsorption and careful removal of MNPs. As illustrated in Figure 7, the concentration of dissolved iron remains under the detection limit of ICP measurement even at a high equilibrium concentration of CSA, while a significant amount of iron was dissolved in the presence of citric acid (CA) as data recalled from our previous paper [22]. Based on these results, we can state that the CSA layer protects the surface of magnetite, and hinders not only corrosion, but also oxidation of MNP, as is also proven by XPS data. Therefore, CSA guarantees good chemical stability for MNP. 

### 2.7. Testing Toxicity of CSA@MNP 

While the low toxicity of negatively charged nanoparticles such as CSA@MNPs in the wide pH range is well-known, it is worth investigating it in vitro [5]. The MTT assays were selected using a human cancer cell line. The results of the cytotoxicity tests are seen in Figure 8. The extent of proliferation inhibition is insignificant so the CSA@MNPs exerted no substantial action on the growth of A431 human cancer cells. The MTT method applied here is widely used for qualifying potential anticancer agents. While the performed MTT assay cannot be a substitute for comprehensive preclinical toxicological evaluation, it indicates that the CSA@MNP preparation is probably non-toxic, because cell killing or growth inhibitory action was not detected.

## 3. Materials and Methods 

### 3.1. Materials

The magnetite nanoparticles were synthesized by co-precipitation, as previously published [19,21,22], the FeCl_2_, FeCl_3_ salts, and NaOH were analytical grade reagents obtained from Molar, Hungary. The synthesized iron oxide was identified as magnetite based on the X-ray diffraction pattern (JCPDS database [77]), characteristic black color, and strong magnetism. The mean diameter of the MNPs determined by transmission electron microscopy, and from the broadening of the most intensive peak of the XRD pattern by using the Scherrer equation was ~10 nm [22]. 

The sodium-salt of chondroitin-sulfate-A (Na_2_CSA) was purchased from Sigma-Aldrich (pristine Na_2_CSA). A small amount of chondroitin-sulfate-C can be found in this material. The pristine Na_2_CSA were used for the surface modification of the MNPs. In this article, the notation "CSA" is used for the sodium–salt, ignoring the actual degree of dissociation of the carboxylic groups. The amount of CSA is expressed through the mole of repeating di-saccharide units. 

NaCl, HCl, and NaOH, analytical grade products of Molar (Hungary) were used to adjust the pH and salt concentration in all experiments. Chondroitinase ABC from Proteus vulgaris and bovine serum albumin were purchased from Sigma Aldrich, sodium-acetate, and tris-(hydroxymethyl)-aminomethane were purchased from Reanal for enzymatic digestion of CSA. Ultra-pure water from a Milli-Q RG water purification system (Millipore) was used. All measurements were performed at 25 ± 1 °C except the enzymatic digestion of CSA and the incubation of cells for cytotoxicity experiments, which were performed at 37 ± 1 °C.

### 3.2. Adsorption Experiment 

The adsorption isotherm of CSA at the MNP surface was determined at pH = 6.3 ± 0.3 and 10 mM NaCl. We used the batch method, in which the MNPs were equilibrated for 24 hours with CSA solutions of concentration between 0.0 and 10 mM at a solid/liquid ratio of 10 g/L. The pH was set at the start of the adsorption. The solid phase was separated by centrifuging at 14,500 rpm for 1 hour (Eppendorf, MiniSpin plus, Germany). 

The equilibrium concentration of the supernatants was determined by two different methods. First, the higher equilibrium concentrations were determined by enzymatic digestion (chondroitinase ABC from Proteus vulgaris, Sigma Aldrich, Germany) according to the enzymatic assay of chondroitinase ABC (Enzyme Commission Number 4.2.2.4) [78] followed by the absorbance measurement at 232 nm in an USB4000 spectrometer (Ocean Optics). The lower equilibrium concentrations were determined by density measurements (DMA58, Anton Paar, Austria). 

The adsorbed amount of CSA (n^σ^_CSA_) was calculated using the material balance equation for adsorption, n^σ^_CSA_ = (V/m)·Δc_CSA_, where V/m is the solution/adsorbent phase ratio (L/g) and Δc_CSA_ is the change in the CSA concentration in the aqueous phase due to the adsorption (mmol/L). The adsorption isotherm was plotted in the function of the equilibrium CSA concentration.

### 3.3. Transmission Electron Microscopy (TEM) 

A Philips CM-10 transmission electron microscope supplied with a Megaview-II camera was used to take the TEM micrographs of magnetite nanoparticles loaded with 0.4 mmol/g CSA. The accelerating voltage of 100 kV was applied; the maximum resolution of the instrument is 0.2 nm. One drop of the highly diluted sol was dried on to a Formwar-coated copper. The average size distribution was determined by evaluating 100 particles using the JMicroVision 1.2.7 software.

### 3.4. Magnetic Measurement (VSM) 

A vibrating sample magnetometer VSM 880 (DMS/ADE Technologies-USA) was used to measure the magnetization curve of CSA@MNP particles with 0.4 mmol/g CSA-loading. The experiment was taken at the RCESCF-UP Timisoara, and the analysis was performed at room temperature on stable sol at 1% by weight, which means that 100 g magnetic fluid contains 1 g magnetite (without CSA). The maximum of the applied field was ~12000 Oe. After the measurement, the value of specific magnetization was calculated by using the exact amount of MNP without CSA and medium. The saturation magnetization (M_S_) value characteristic for the magnetite nanoparticles was calculated by extrapolation to an infinite field in an M vs. 1/H^2^ plot [63].

### 3.5. Infrared Spectroscopy (FTIR-ATR) 

FTIR-ATR spectra were recorded with a Bio-Rad Digilab Division FTS-65A/896 spectrometer with DTGS detector, using a Harrick’s Meridian Split Pea Diamond ATR accessory. The absorbance of the samples was measured in single reflection mode over the 400−4000 cm^−1^ range (with resolution of 2 cm^−1^), accumulating 256 scans. Magnetite dispersion, aqueous CSA solution, and CSA@MNP suspension were dried on the crystal surface. The CSA@MNP was loaded with 0.2 mmol/g CSA, the pH of all samples was set to 6.3 ± 0.3, and the ionic strength was 10 mM (NaCl). The background spectra were measured on clean and dry diamond crystal.

### 3.6. X-ray Photoelectron Spectroscopy (XPS) 

An XPS spectrometer SPECS equipped with an Al/Mg dual-anode X-ray source, a PHOIBOS 150 2D CCD hemispherical energy analyzer, and a multichanneltron detector with a vacuum maintained at 1 × 10^−9^ Torr was used to record the XPS spectra. The Al K_α_ X-ray source (1486.6 eV) was operated at 200 W. The XPS survey spectra were recorded at 30 eV pass energy and 0.5 eV/step. The high-resolution spectra for the individual elements (Fe, C, O, N, and S) were recorded by accumulating 10 scans at 30 eV pass energy and 0.1 eV/step. MNP nanoparticles loaded with 0.2 mmol/g CSA were used for the experiment. The suspension was prepared at pH = 6.3 ± 0.3 and 10 mM NaCl. The CSA@MNP particles were dried on an indium foil, and the surface of the samples was cleaned by argon ion bombardment (300 V). The spectra were recorded before and after the cleaning. Data analysis and curve fitting was performed using CasaXPS software with a Gaussian–Lorentzian product function and a nonlinear Shirley background subtraction. The high resolution spectra were deconvoluted into the components corresponding to particular bond types.

### 3.7. Dynamic Light Scattering (DLS) 

The average hydrodynamic diameter (Z-Ave) of MNP and CSA@MNP particles characteristic for their aggregation state were determined at 25 ± 0.1 °C using a Nano ZS apparatus (Malvern, UK) operating in backscattering mode at an angle of 173°. The samples were diluted to give an optimal intensity of ~10^5^ counts per second. To get comparable data, the dispersions were homogenized in an ultrasonic bath for 10 s, after which 2 min relaxation was allowed before the start of the DLS measurements. The influence of the CSA-loadings on the average hydrodynamic diameter of CSA@MNPs was determined at pH = 6.3 ± 0.3 and 10 mM NaCl. The effect of pH variation (between 3 and 10) on the aggregation state of the bare and coated nanoparticles (i.e., at 0.0, 0.05, 0.1, 0.2, and 0.4 mmol/g CSA-loadings) was studied at 10 mM NaCl. We used the second or third-order cumulant fit of the autocorrelation functions, depending on the degree of polydispersity.

### 3.8. Electrophoresis Experiments

Electrophoretic mobilities of the naked MNP and different CSA@MNP nanoparticles were measured in a Nano ZS (Malvern, UK) apparatus with a 4 mW He−Ne laser source (λ = 633 nm) using disposable zeta cells (DTS 1060) at 25 ± 0.1 °C. The zeta-standard of Malvern (−55 ± 5 mV) was used for calibration. The measured samples were the same as those prepared for the DLS studies. The Smoluchowski equation was applied to convert electrophoretic mobilities to electrokinetic potential values. The accuracy of the measurements was ±5 mV.

### 3.9. Coagulation Kinetics Experiments 

The effect of adsorbed CSA on the colloidal stability of magnetite nanoparticles was tested in coagulation kinetics experiments at different NaCl concentrations at pH = 6.3±0.3. The CSA-loadings were 0.1, 0.2, and 1.0 mmol/g. The change in the average hydrodynamic diameter (Z-Ave) was measured by DLS at 25 ± 0.1 °C for 15 min with a resolution of 60 s. The initial slope of the kinetic curves, i.e., dZ-Ave/dt = f(t) is proportional to the coagulation rate [49,72,73], so the stability ratio (W) can be calculated as the ratio of the initial slopes belonging to the fast and slow coagulation, respectively. The salt tolerance of the samples is given as the critical coagulation concentration (CCC) obtained from the stability ratio (log_10_ W) vs. electrolyte concentration (log_10_ c) functions.

### 3.10. Iron Dissolution Experiments 

Chemical stability of the CSA-coated nanoparticles was characterized by the determination of the dissolved iron in the aqueous dispersions of CSA@MNPs after 24 hours adsorption at pH = 6.3 ± 0.3 and 10 mM NaCl. The samples were selected along the adsorption isotherms: Four of them evenly distributed in the rising part and two of them in the plateau region. The equilibrium supernatants were separated at 500 mM NaCl by centrifuging at 13,000 rpm for 1 hour. The iron content was measured in an Agilent 7700x ICP-MS spectrometer. Reliable iron concentration cannot be determined below the limit of ~150 ppb.

### 3.11. Anti-Proliferative Assays 

A431 human cancer cell line (isolated from skin) was maintained in minimal essential medium supplemented with 10% fetal bovine serum (FBS), 1% nonessential amino acids, and an antibiotic-antimycotic mixture. The cells were grown in a humidified atmosphere of 5% CO_2_ at 37 °C. The cell line was purchased from the European Collection of Cell Cultures (Salisbury, UK). Cells were seeded onto 96-well plates at a density of 5000 cells/well and allowed to stand overnight, after which the medium containing the tested agent (CSA@MNP with 0.2 mmol/g CSA-loading) was added. Final MNP concentration in the samples varied between 0.001 and 0.1 mg/mL. After a 72 hours incubation period, viability was determined by the addition of 0.02 mL of MTT ([3-(4,5-dimethylthiazol-2-yl)-2,5-diphenyltetrazolium bromide]) solution (5 mg/mL). The precipitated formazan crystals were solubilized in dimethyl sulfoxide (DMSO) (0.1 mL), and the absorbance was measured at 545 nm with an ELISA (enzyme-linked immunosorbent assay) reader [79]. The samples were characterized by the growth inhibition capacity expressed as a percentage.

The majority of experiments were performed at pH = 6.3 ± 0.3 and 10 mM NaCl, so we simplify the notion of this pH value as ~6.3, and omit to note pH and ionic strength unless it has special significance or the values are different.

## 4. Conclusions

In this work, CSA-coated core-shell magnetite nanoparticles were synthesized in a post-synthesis coating process. We characterized both the mechanism of surface modification and the physical, physicochemical, and colloidal properties of the CSA@MNPs. It turned out that the cetylpyridinium chloride content of CSA affects the stabilization of the magnetite nanoparticles, and so accurate conditions are required to prepare stable CSA@MNP magnetic fluid. TEM images and VSM measurements evidenced that the size and magnetic properties of magnetite cores in CSA@MNP did not change during the post-synthesis coating process. FTIR-ATR spectra and XPS measurements proved that CSA forms inner-sphere metal–carboxylate complexes (≡Fe–OOC–R) on the MNP surface. This bond formation may support the H-type adsorption isotherm, and it enables the preparation of magnetic fluid with low equilibrium CSA concentration favorable in later in vivo applications. The CSA@MNPs at loading above 0.2 mmol/g are electrosterically stabilized over the wide range of pH (>4, alike in numerous biological media). The salt tolerance of CSA@MNP is sufficiently high to prevent particle aggregation in physiological medium. There was no iron leaching and oxidation of magnetic core owing to the excellent protective effect of the CSA layer on MNP particles. The noteworthy chemical stability of CSA-coated MNPs, the outstanding pH and salt tolerance of the MFs accompanied by their non-toxicity based on the MTT assays support the feasibility of chondroitin-sulfate-A-coated magnetite nanoparticles in biomedical applications.

Obviously, further in vitro experiments are required to test the prepared magnetic fluid (e.g., erythrocyte sedimentation rate measurements, blood smear tests, and blood cell viability experiments to characterize the hemocompatibility), but the CSA@MNPs are favorable candidates for biomedical applications, not only as agents in MRI diagnostics [33,45] and anticancer drug carriers [38], but also that in magnetic hyperthermia; even the theranostic application of CSA@MNP products may be feasible. Furthermore, these CSA@MNP particles are suitable for preparation of biocompatible magnetic hyaluronate-gel for a potential application as intra-articular injections in the treatment of osteoarthritis.

## Figures and Tables

**Figure 1 ijms-20-04096-f001:**
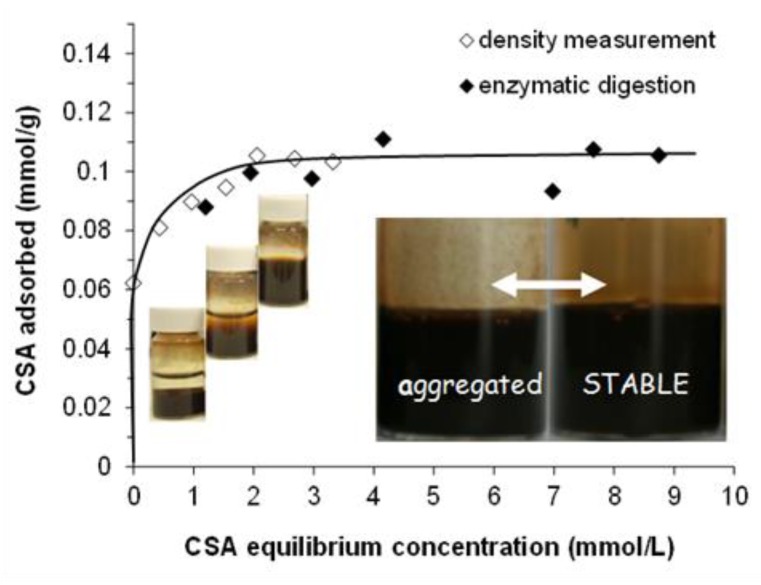
Adsorption isotherm of chondroitin-sulfate-A (CSA) on magnetite nanoparticles (MNP) at pH ~6.3 and 10 mM NaCl. With increasing CSA concentration, the colloidal state of samples changes characteristically from aggregated to stable, as seen in the vials. Stable CSA@MNP (see the inserted photos) can only be prepared under accurate conditions. (The amount of CSA is expressed through the number of repeating units in mmol. The line is drawn as a guide for the eye).

**Figure 2 ijms-20-04096-f002:**
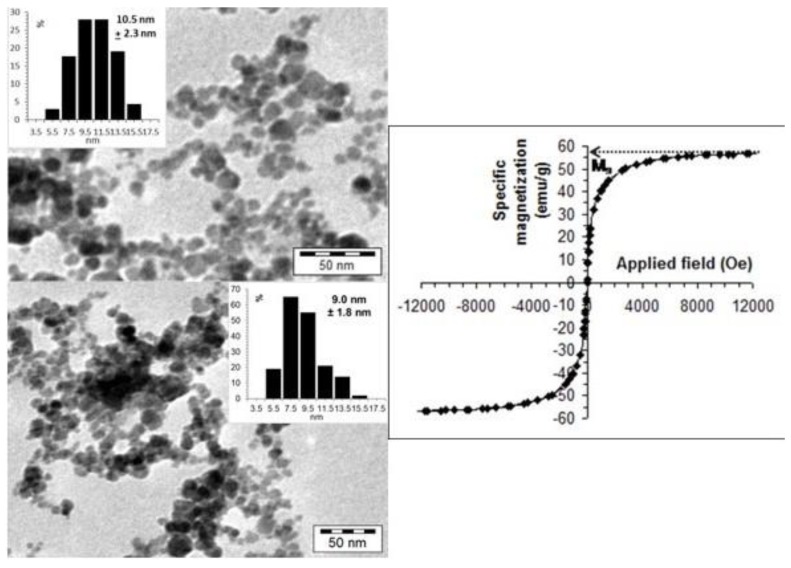
(**a**) TEM picture and particle size distribution of CSA@MNP nanoparticles at pH ~6.3 (top) and that of naked MNPs (bottom); (**b**) specific magnetization curve measured in the dispersion CSA@MNPs at room temperature (10 g/L in water at pH ~6.3). (The M_s_ is assigned only for illustration.).

**Figure 3 ijms-20-04096-f003:**
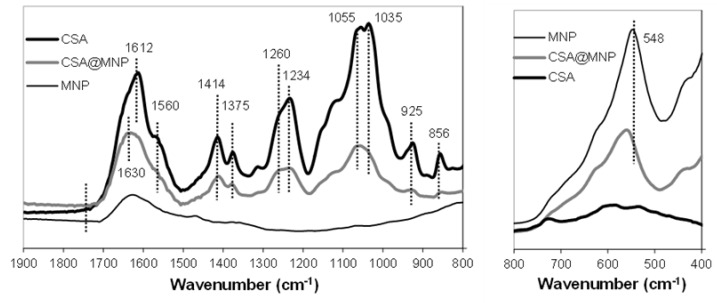
FTIR-ATR spectra (absorbance is given in arbitrary units) of CSA, magnetite (MNP), and CSA-coated magnetite (CSA@MNP) in the 800−1900 cm−1 (left side) and in the 400−800 cm^−1^ (right side) range. Samples were dried on the diamond crystal from solution/dispersions at pH ~6.3.

**Figure 4 ijms-20-04096-f004:**
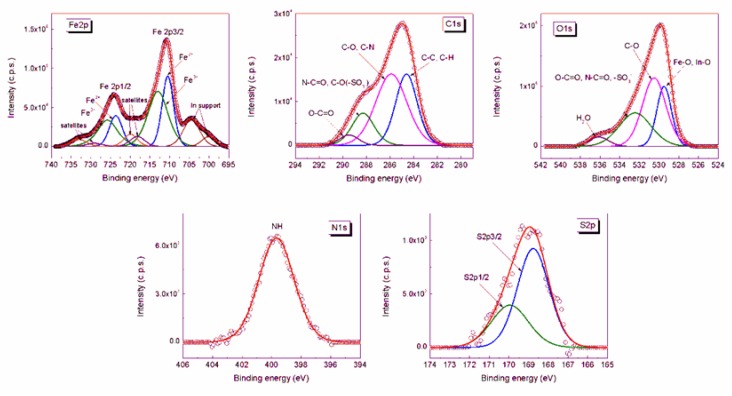
XPS spectra of Fe 2p, C 1s, O 1s, N 1s, and S 2p core levels from CSA@MNPs sample at pH ~6.3 and 10 mM NaCl.

**Figure 5 ijms-20-04096-f005:**
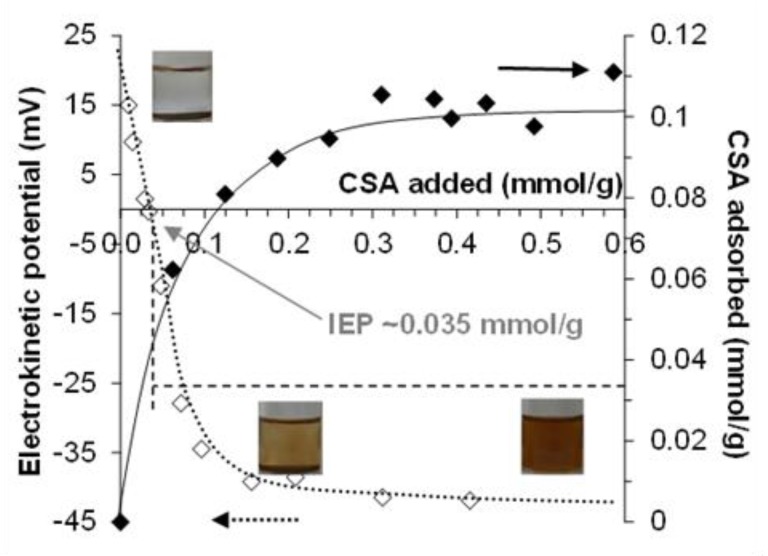
Effect of CSA-addition on the charge state of MNPs and the colloidal stability of the dispersions (inserted photos) at pH ~6.3 and 10 mM NaCl. The adsorption and the electrokinetic potential data are shown. (Lines are drawn as guides for the eye).

**Figure 6 ijms-20-04096-f006:**
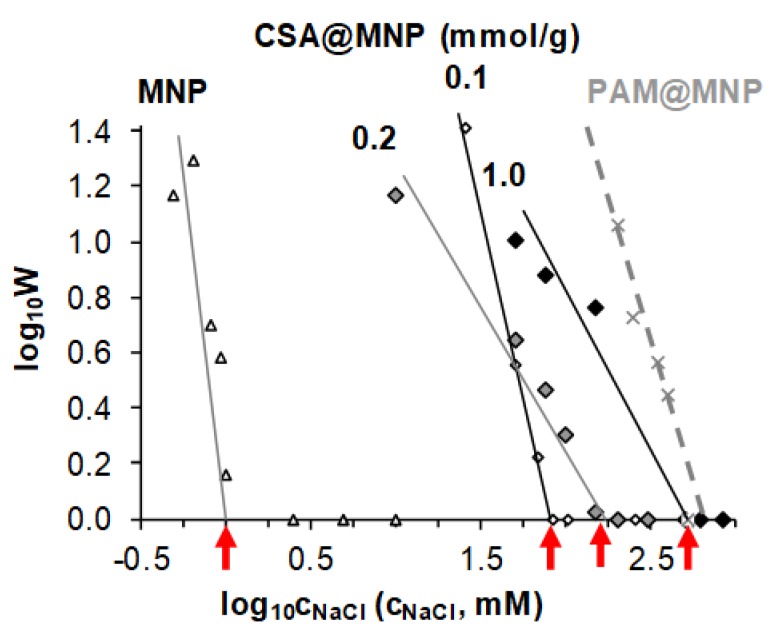
Stability plot to determine the values of the critical coagulation concentration (marked with red arrows) of MNPs in the absence of CSA coating (open triangle symbols) and in the presence of 0.1, 0.2, and 1.0 mmol/g CSA (diamond symbols), measured at pH ~6.3. The previous results of PAM-coated MNPS with 1.2 mmol COOH/g MNP loading [19] (dashed line) are recalled here for comparison.

**Figure 7 ijms-20-04096-f007:**
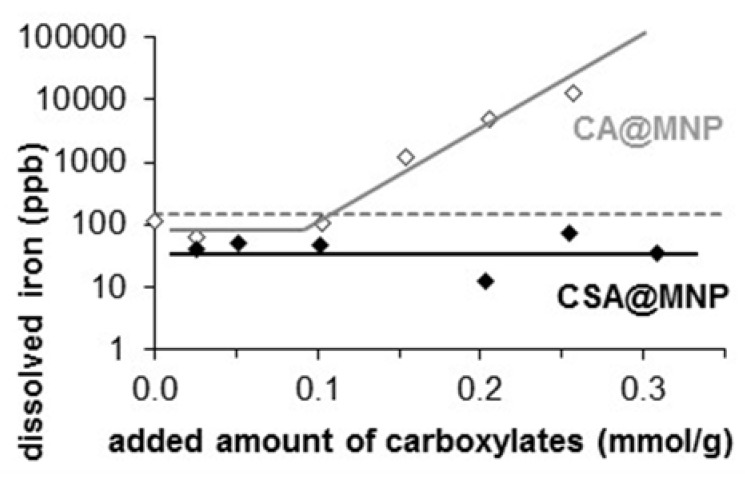
Dissolved iron concentration in CA@MNP and CSA@MNP dispersions (in 10 mM NaCl solution at pH ~6.3) in the function of the added amount of carboxylates, such as citric acid (CA) [22] and CSA. (The lines are drawn to CA and CSA data to guide the eyes. The dotted line indicates the detection limit of iron).

**Figure 8 ijms-20-04096-f008:**
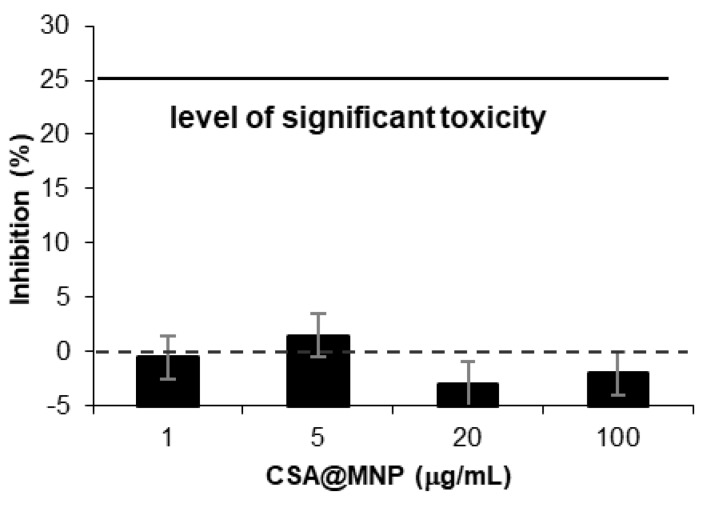
Cytotoxicity experiments: growth inhibition capacity of CSA@MNP (at 0.2 mmol/g CSA-loading) on A431 cell line.

**Table 1 ijms-20-04096-t001:** Characteristic IR bands and their shifts (cm^−1^) due to the CSA-adsorption on magnetite nanoparticles at pH ~6.3 and 10 mM NaCl.

	MNP	CSA	CSA@MNP	Δν *
Fe–O (≡Fe–OH)	548		559	11
C = O (–COOH)		-	-	-
C–O (sym., –COO^−^)		1375	1379	4
C–O (asym., –COO^−^)		1612	1630	18
C–O–S (–O–SO_3_^−^)		856	856	0
S = O (–O–SO_3_^−^)		1260	1260	0

* Δν = ν_adsorbed_−ν_free_.

**Table 2 ijms-20-04096-t002:** Assignment of peaks in the XPS spectra of the CSA@MNPs together with binding energies (position, eV), peak full width at half maximum (fwhm, eV), and atomic concentration percentages (atomic conc, (%).

Peak Name	Position (eV)	fwhm (eV)	Atomic Conc (%)
Fe^2+^ 2p3/2	710.46	3.410	5.629
Fe^3+^ 2p3/2	712.98	6.000	7.950
Fe^2+^ 2p1/2	723.66	3.882	5.450
Fe^3+^ 2p1/2	725.78	6.000	7.697
Fe^3+^ satellite 2p3/2	719.92	4.075	1.184
Fe^2+^ satellite 2p3/2	718.19	4.374	1.038
Fe^3+^ satellite 2p1/2	733.06	4.627	1.507
Fe^2+^ satellite 2p1/2	729.03	4.649	0.690
C 1s; C‒C, C‒H	284.61	2.147	6.221
C 1s; C‒O, C‒N	285.90	3.121	9.037
C 1s; N‒C = O, C‒O(‒SO_3_^‒^)	288.29	2.218	2.935
C 1s; O‒C = O	289.48	1.929	0.794
O 1s; Fe‒O, In‒O	529.49	1.746	11.066
O 1s; C‒O	530.51	2.802	20.488
O 1s; O‒C = O, N‒C = O, ‒SO_3_^‒^	532.43	3.838	13.709
O 1s; H_2_O	536.11	2.356	2.342
N 1s; N‒H	399.67	2.814	1.717
S 2p3/2; ‒SO_3_^‒^	169.95	2.230	0.271
S 2p1/2; ‒SO_3_^‒^	168.77	1.909	0.277

**Table 3 ijms-20-04096-t003:** Effect of the CSA-loading on the isoelectric point (IEP) and the pH-range of MNPs’ aggregation at 10 mM NaCl.

CSA-Loading (mmol/g)	pH of IEP	pH-Range of Aggregation
0.00	~8	~5 ‒ ~10
0.05	~6	~3 ‒ ~10
0.10	~4	< ~5
0.20	< 3	< ~3.5
0.40	< 3	< ~3

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
