# Peer review of "Chondroitin-Sulfate-A-Coated Magnetite Nanoparticles: Synthesis, Characterization and Testing to Predict Their Colloidal Behavior in Biological Milieu"

_ijms, 2019, doi:10.3390/ijms20174096_

Round 1

Reviewer 1 Report

Dear authors,

Thanks for the interesting reporting on "Chondroitin-sulfate-A-coated magnetite nanoparticles: synthesis, characterization and testing to predict their colloidal behavior in biological milieu" where you describe pH-dependent electrical potential upon coating CSA on magnetic nanoparticles, presence of the material of the coated nanocomposites, and its biocompatibility.

Several questions regarding this work are the following,   

1. is the coating material, CSA, chosen for any potential biological applications? any functionality may be involved?

2. At line 340, the author said that the range of stability is at pH below 5 or above 10, but in table 3, it seems to describe a different range.

3. It is know that cancer cells or tumor has higher acidity than normal cells or tissue, since the author used cancer cells for biocompatibility assay, a description on the suitability of the charge density of the CSA-coated MNPs for normal and cancer cells may be helpful to justify its biocompatibility and consequences of its application.

4. Author present different kinds of material characterization, pH, electric potential, structure, chemical contents, what are the correlation between these characteristic properties and biological application? A description of these can be very helpful to readers who may not work in the respective fields.

Reviewer 2 Report

Dear Editor, dear authors,

The manuscript of Tóth et al. describes the formation of chondroitin sulfate A (CSA)-coated magnetite nanoparticles (MNPs), the characterization of these CSA-coated MNPs using various physicochemical methods and finally an assessment of the salt stability and cytotoxicity of these nanoparticles. The subject is of high relevance, as nm-sized magnetite particles typically exhibit superparamagnetism, an physical phenomenon which allows for various interesting applications of such nanoparticles like their use in hyperthermal treatment of cancer cells or as contrast agents in magnetic resonance imaging. The manuscript covers an interesting topic and reads well but became quite lengthy as, in my opinion, many details have been given in the main part of the manuscript that could, in principle, be given instead as supplementary material, allowing to shorten the main manuscript (by referring to the supplementary material where necessary). Furthermore, some details are not clearly described (see items below) so that I cannot yet recommend publication of the manuscript in the International Journal of Molecular Sciences, but may do so if the authors properly addressed the following issues.

MAJOR REVISION

1. The authors mention the necessity to convert pristine sodium chondroitin sulfate A, Na2CSA, into H2CSA. Unfortunately, I can neither fully follow the incentive of the procedure nor the related characterizations shown in the manuscript.

1a. According to the methods section of the manuscript, H2CSA is formed from Na2CSA by adjusting the pH-value of the Na2CSA solution to pH = 1, followed by dialysis against ultra-pure water. As only the COOH-groups protonate under these conditions (the authors write themselves that the pKa of the sulfate groups is so low that this group is always deprotonated at all pH-values used in the manuscript; see e.g. page 5, line 157) I assume that the authors aim to remove sodium ions from the COO--groups? Even if this procedure leads to the formation H2CSA, a claim which should be supported e.g. by an elemental analysis of the product as I would expect sodium to remain at the sulfate groups, it will most likely convert back to Na2CSA during the experiments, as both the sulfate and COOH-groups are fully deprotonated at pH ~ 6.3 (used in the manuscript) and as most (if not all) experiments are carried out in presence of at least 10 mM sodium chloride, right?

1b. In this context, I do not understand, why the proton uptake shown in Fig. 2 differs between Na2CSA and H2CSA in particular at high sodium chloride concentrations, at which I would expect binding of sodium ions to the deprotonated COOH-groups, i.e., conversion of H2CSA back to Na2CSA?

1c. The formation of H2CSA rather seems to be an ion exchange than a true synthesis procedure (as e.g. claimed on page 3, line 107).

2. The items above show my difficulties in understanding that Na2CSA has to be converted to H2CSA to allow for reaching a well-defined initial state (as claimed on page 3, line 119), in particular as I would expect reformation of Na2CSA from H2CSA at the experimental conditions. How would e.g. the CSA adsorption data or MNP stability data look like if Na2CSA would be used instead of H2CSA?

3. On page 6, line 218, the authors refer to the TEM image in Fig. 5a and write that the CSA layer prevents MNP aggregation, although the TEM image rather shows aggregated MNPs instead of isolated nanoparticles. How can this conclusion be drawn from Fig. 5a?

4. If I understand it correctly, Fig. 9 shows the common logarithm of the ratio of a slow and fast aggregation process of MNPs monitored using DLS. This means that using DLS is used to determine the time-dependent size distribution of MNP samples (under investigation) and that these time-dependent size distributions show a slow and fast coagulation process, i.e., an increase in MNP size which has a slow and a fast time constant, right? If so, how meaningful is it to use Fig. 9 to determine the stability of the MNP samples? If there is a slow coagulation process, the MNPs will still aggregate after a certain time even if subjected to salt conditions well below the so-called critical coagulation concentration, right?

MINOR ISSUES

1. I assume that the CSA concentrations given in the manuscript always refer to the molar concentration of the repeating units? Also the degree of sulfation mentioned refers to the number of sulfates per repeating unit, i.e., per di-saccharide, rigth?

2. The authors do not give any details on the co-precipitation procedure used for MNP generation, e.g., which concentration values were used, which conditions were necessary to perform the co-precipitation procedure etc.?

3. Fig. 11, MTT test: Which control measurements were performed? We typically complement such measurements with a negative and positive control using buffer and SDS, respectively, in order to show that the assay properly reports cell death.

4. Page 13, line 404: “Scherrer equation was ~10 nm”. What does this mean; that the X-ray diffraction (XRD) peak with leads to crystallite sizes of ~ 10 nm? Furthermore, does the XRD pattern indicates magnetite as sole crystallographic phase of the synthesized MNPs?

5. Page 15, line 490: Is it 105 (as written) or 105 counts?

Round 2

Reviewer 2 Report

Dear Editor, dear authors,

The manuscript of Tóth et al. describes the formation of chondroitin sulfate A (CSA)-coated magnetite nanoparticles (MNPs), the characterization of these CSA-coated MNPs using various physicochemical methods, and an assessment of the salt stability and cytotoxicity of these nanoparticles. The subject is of high relevance, as nm-sized magnetite particles typically exhibit superparamagnetism, a physical phenomenon which allows for various interesting applications of such nanoparticles like their use in hyperthermal treatment of cancer cells or as contrast agents in magnetic resonance imaging. The manuscript covers an interesting topic, reads very well, and the authors have addressed almost all of my concerns, so that I would like to recommend publication of the manuscript in the International Journal of Molecular Sciences after some minor issues have been addressed by the authors.

MINOR REVISION

Regarding my major concern 3: I expressed my confusion about the authors’ statement in the main text that the “…CSA layer prevents the close aggregation of the nanoparticles…” while Fig. 5a, which was referred to in this statement, appears to show aggregated nanoparticles. The authors’ response was “In fact, “... the close aggregation of the nanoparticles…” is written.”, which is still confusing as in fact “… CSA layer prevents the close aggregation of the nanoparticles…” is written in both versions of the manuscript. In my opinion, this cannot be concluded from Fig. 5a, as I’m not able to resolve isolated nanoparticles there. The authors claimed in their response that “When samples are dried onto the TEM grid, particles aggregate seriously; however, several white gaps can be identified between the MNPs in Fig. 5a.” To uphold this claim, it is necessary to compare TEM images of bare and CSA-coated MNPs, as only in this way it will become clear, if the white gaps (which I’m not able to resolve clearly) are significant and support the claim.

The description of the coagulation kinetics is still slightly misleading as the authors define the stability ratio using a so-called fast and slow coagulation rate without giving further details on these 2 rates. This is, in my opinion, better done in the authors’ answers to my major concern 4, in which it becomes clear that they measure the coagulation rate in dependence of the salt concentration, which has a small value at sufficiently small salt concentrations and increases with increasing salt concentration until reaching a (faster) saturation value. The stability ratio is then given by the saturation coagulation rate divided by the coagulation rate at given salt concentration (which is a more exact definition than used in the manuscript, as the so-called slow coagulation rate is identical to the so-called fast coagulation rate above the CCC, right?). Nevertheless, as the stability is calculated based on the saturation coagulation rate, I would suggest to add the value of this rate to Fig. 6 (e.g., in the caption) to allow for a fair comparison of the MNPs shown (the 1.0 CSA@MNPs have a better CCC than the 0.1 CSA@MNPs but could have, in principle, a faster saturation coagulation rate, leading to comparable coagulation rates below the CCC, right?).
